# IFN-α/β Signaling Is Required for CDG-Mediated CTL Generation and B Lymphocyte Activation

**DOI:** 10.3390/pharmaceutics14122821

**Published:** 2022-12-16

**Authors:** Ahmed E. I. Hamouda, Kai Schulze, Thomas Ebensen, Carlos Alberto Guzmán, Darío Lirussi

**Affiliations:** 1Department of Vaccinology and Applied Microbiology, Helmholtz Centre for Infection Research, 38124 Braunschweig, Germany; 2Currently at Lab of Dendritic Cell Biology and Cancer Immunotherapy, VIB Center for Inflammation Research, 1050 Brussels, Belgium; 3Instituto Nacional de Medicina Tropical (INMeT), Administración Nacional de Laboratorios e Institutos de Salud “Dr Carlos Malbrán”, Puerto Iguazú 3370, Misiones, Argentina

**Keywords:** adjuvant, immunomodulatory, vaccines, IFN-α/β, cyclic di-nucleotides

## Abstract

Among cyclic di-nucleotides (CDN), both cyclic di-AMP (CDA) and di-GMP (CDG) are promising adjuvants and immune modulators. These molecules are not only able to induce profuse antibody production but also predominant T helper 1 and cytotoxic CD8 T lymphocytes (CTL) responses, which enable their use for vaccination against intracellular pathogens as well as in cancer immunotherapy. However, for their successful translation into the clinic, a comprehensive understanding of CDN mode of action is still essential. Consistent with evidence in the literature, we show here that IFN-α/β (Type I IFN) is crucial for CDG-mediated B cell activation. We recently determined the key role of type I IFN signaling for CDA-mediated enhancement of immunogenicity. Based on the biological activities of type I IFN, in this study, we hypothesized that it might also be required for CTL induction by CDG. We disclose here the mode of action of type I IFN signaling in CDG-mediated cross-presentation and subsequent CTL generation.

## 1. Introduction

Among novel adjuvant or immunostimulatory molecules being evaluated in clinical trials, cyclic di-nucleotides (CDN) such as cyclic di-AMP (CDA) and cyclic di-GMP (CDG) are promising candidate immune modulators [1]. These adjuvants are not only able to induce profuse antibody production but also predominant T helper 1 (Th1) and cytotoxic CD8 T lymphocytes (CTL) responses [2]. These characteristics enable us to exploit them for vaccination against intracellular pathogens as well as in cancer immunotherapy [3,4]. In the case of CDA, it has been recently shown to trigger a stronger antibody, Th1 signaling and IFN-γ production than other adjuvants with well-defined molecular targets when co-formulated with soluble antigens used by different routes [5]. Nevertheless, a comprehensive understanding of CDN mode of action is a prerequisite for their successful development and their translation into the clinic. Since we recently disclosed the key role of IFN-α/β signaling for CDA-mediated immunogenicity [6], we hypothesized that IFN-α/β must also be essential for CDG-mediated immune responses. Thus, we investigated the effect of CDG on bone marrow-derived dendritic cells’ (BMDC) cross-presentation capacity as well as the expression of co-stimulatory molecules and whether it is dependent on type I IFN signaling. Similar to CDA and CDG-triggered cross-presentation, the subsequent cross-priming and resultant CTL generation are dependent on type I IFN signaling. It was previously reported that the loss of type I IFN signaling diminishes humoral responses. In this study, we evaluated the activation of B cells in WT and IFNAR KO (*IFNAR1^−/−^*) mice immunized with CDG. We show here that the lack of type I IFN signaling negatively affected the numbers and activation status of B cells. Therefore, our data strongly indicate that type I IFN signaling is also necessary for CDG-mediated humoral and cellular immune responses.

## 2. Materials and Methods

### 2.1. Animals

C57BL/6 wild type (WT) and OT-I mice were purchased from Harlan (Rossdorf, Germany) and bred at the Helmholtz Center for Infection Research (HZI), respectively. *IFNAR1^−/−^* (IFN receptor KO mice) were bred at the HZI. TNFR1a/b^−/−^ (B6.129S-Tnfrsf1a^tm1lmx^ Tnfrsf1b^tm1lmx^/J) [7] were purchased from Jackson Laboratory (Bar Harbor, ME, USA). All C57BL/6 animals were kept under specific pathogen-free conditions. Animals were handled in accordance with the German animal protection act (TierSchG BGB1. I S 1666; 18.07.2016). The discussed animal experiments were authorized by the local authorities (Consumer Protection and Food Safety, LAVES) of Lower Saxony (Germany) under permit number 33.4-42502-04-13/1281. We immunized WT and *IFNAR1^−/−^* with the model antigen ovalbumin (OVA; Hyglos GmbH, Baden-Wurttemberg, Germany) and CDN (7.5 µg/dose subcutaneously) and measured cytokine production by ELISPOT and IgG by ELISA. The cellular immunity of the *cytotoxic* T lymphocytes (CTL) was measured after vaccination by a CTL assay, which is described in [8]. CTL proliferation was measured by flow cytometry after the passive transfers of CFSE-stained CD8 OT-I T cells according to routine procedures in our institutional labs [9].

### 2.2. ELISPOT

Splenocytes IFN-γ secretion was measured by ELISPOT using 96-well plates with a hydrophobic High Protein Binding Immobilon-P-Membrane (BD Pharmingen) or MultiScreen HTS filter plates (Millipore, Darmstadt, Germany) were coated with anti-IFN-γ antibodies diluted in PBS and incubated for 18 h at 4 °C. After 2 h of blocking of nonspecific binding sites at RT using 200 μL/well of complete medium, 4 × 10^5^ and 2 × 10^5^ spleen cells/well were added and incubated (5% CO_2_ in complete RPMI medium) in the absence (blank, only complete RPMI medium added) or presence of the MHC-I immunodominant peptide SIINFEKL (amino acids 257–264) and the MHC-II immunodominant peptide ISQAVHAAHAEINEAGR (amino acids 323–339) of OVA [5 µg/mL], respectively. A total of 5 μg/mL of mitogen Concanavalin A (Sigma, Burlington, MA, USA) was used as a positive control. After 16 h of incubation at 37 °C, plates were washed and incubated for 2 h at RT in the presence of biotinylated detection antibodies (mouse-specific ELISPOT pair antibodies -BD Bioscience, BD Pharmingen-), according to the manufacturer’s instructions. After thorough washing, plates were incubated for 1 h at RT in the presence of peroxidase-conjugated streptavidin. After an additional washing step, cytokine-secreting cells were detected by a color reaction by adding AEC substrate (diluted in 0.1 M acetate buffer pH 5.0) containing 0.05% H_2_O_2_ (30%) to the cells. The reaction was then stopped using distilled water as soon as spots developed. The images were acquired using an ELISPOT scanner (Cellular Technology, Ltd. -CTL-, Bonn, Germany) and the colored spots were quantified by ImmunoSpot image analyzer software v3.2 (CTL, Bonn, Germany). Spot numbers without re-stimulation were subtracted from re-stimulated ones for normalization purposes. The number of spots per group was obtained by pooling replicates or the average value of replicates was calculated for each group, and then expressed in spots/millions of cells. Wells with backgrounds higher than 5% were not recorded. The displayed averages and errors are from two cell concentrations evaluated for each treatment. The results are representative of two independent vaccination experiments.

### 2.3. ELISA

OVA-specific IgG antibody titers in sera of mice vaccinated with CDN + OVA, OVA, or vehicle were sampled according to the diagram in Figure 5A. ELISA was performed by coating high-binding protein plates with OVA protein (2 µg/mL in 0.05 M carbonate buffer) for 18 h at 4 °C. Plates were blocked for 2 h with 3% bovine serum albumin (BSA) in PBS. Serial 2-fold dilutions of sera in 3% BSA/PBS were added (100 µL/well) and incubated for 1 h at 37 °C. Plates were washed six times using 1% BSA/0.05% Tween 20 in PBS and biotinylated goat anti-mouse IgG (Sigma, Burlington, MA, USA) were added (1:5000 in washing buffer) for 1h at 37 °C. The 96 well plates were washed again and incubated for 1 h at RT in the presence of peroxidase-conjugated streptavidin (BD Pharmingen, San Diego, CA, USA). The plates were then thoroughly washed and developed with ABTS [2,20-azino-bis(3-ethylbenzthiazoline-6-sulfonic acid)] in a 0.1 M citrate-phosphate buffer (pH 4.35) containing 0.01% H_2_O_2_. The color reaction was monitored at different time points (5, 15 and 30 min). Endpoint IgG titers are expressed as absolute values of the last dilution, giving an optical density (OD405 nm) two times higher than the values of the negative control (blank) after 30 min of incubation. Results are representative of four independent vaccination experiments.

### 2.4. Generation of Bone Marrow-Derived Dendritic Cells (BMDC)

The bone marrow was flushed out of the euthanized mouse’s femurs and tibias with RPMI 1640. Following tissue disruption and red blood cells lysis, the bone marrow cells were cultured in RMPI 1640 medium supplemented with 50 U/mL penicillin, 50 µg/mL streptomycin, 50 µg/mL gentamycin, 10% *v*/*v* fetal calf serum (FBS; all from Gibco, Grand Island, NY, USA), 2 mM L-glutamine (Hyglos GmbH, Bernried, Germany) and 20 ng/mL granulocyte-macrophage colony-stimulating factor (GM-CSF; Biolegend, San Diego, CA, USA). The medium was partially replenished every day. BMDC were obtained after 7 days of differentiation. These cells were then used as antigen presenting cells (APC) by treating them with 5 µg/mL endotoxin-free OVA in the presence or absence of 5 µg/mL CDA/CDG (InvivoGen, San Diego, CA, USA) or 1 µg/mL lipopolysaccharide (LPS, InvivoGen, San Diego, CA, USA). To elucidate the cross-presentation pathway, BMDC were alternatively treated with 100 nM MG-132, 10 µM lactacystin or 10 µM leupeptin (all from Sigma-Aldrich, Darmstadt, Germany).

### 2.5. Flow Cytometry

Spleens and lymph nodes (LN) were mechanically dissociated against 100 µm cell strainers (BD Falcon, Franklin Lakes, NJ, USA) to obtain a single cell suspension. Following RBC lysis, cells were stained with fluorochrome-conjugated antibodies for 30 min at 4 °C. Antibodies used to stain mouse CD11c (clone N418, Allophycocyanin-conjugated) and mouse CD40 (clone Mrz23, allophycocyanin-conjugated) were purchased from eBioscience, San Diego, CA, USA. Antibodies used to stain mouse CD11c (clone N418, BV785-conjugated), mouse SIINFEKL peptide bound to H-2Kb MHC-I (clone 25-D1.16, PE-conjugated), and mouse CD86 (clone GL1, BV605-conjugated) were purchased from BioLegend, San Diego, CA, USA. Anti-mouse CD80 (clone 16-10A1, PE-conjugated) was purchased from BD, USA. Data were acquired using LSR Fortessa (BD Biosciences, San Jose, CA, USA) and FACSDiva 6.2 (BD Biosciences, USA). The acquired data were analyzed using FlowJo (FlowJo, LLC, Ashland, OR, USA).

### 2.6. In Vivo OT-I CD8+ T Cells Proliferation Assay

CD8^+^ OT-I T cells were purified using Miltenyi CD8a (Ly-2) MicroBeads and LS columns (Miltenyi, Bergisch Gladbach, Germany) according to the manufacturer’s instructions. Purified CD8^+^ OT-I cells were stained with a 3.5 µM carboxyfluorescein succinimidyl ester (CFSE; Molecular Probes, Eugene, OR, USA) for 10 min at 37 °C. 2 × 10^6^ CFSE-labeled CD8^+^ OT-I T cells were administered intravenously (i.v.) through the tail vein of WT and IFNAR1^−/−^ mice. After 16 h, mice were subcutaneously (s.c.) immunized with 50 µg OVA +/− 7.5 µg CDG. After 3 days, spleens and LN were collected and analyzed by flow cytometry. The proliferation of CD8^+^ OT-I T cells was proportional to the decrease in the intensity of CFSE.

### 2.7. In Vivo CTL Assay

Splenocytes were pulsed with either 1 µM SIINFEKL peptide, 1 mg/mL OVA protein or vehicle as a negative control for 90 min at 37 °C. Splenocytes pulsed with SIINFEKL, OVA and vehicle were further stained with 0.1 µM (low), 0.5 µM (medium) and 5 µM (high) CFSE, respectively. Cells were mixed in a 1:1:1 ratio and 3 × 10^7^ cells/mouse were intravenously administered. 18 h later, spleens and LN were collected from the recipient mice and analyzed by flow cytometry. The specific lysis of SIINFEKL- and OVA-loaded target cells was calculated as a ratio to unpulsed target cells: ratio = percentage of unpulsed CFSE_high_ cells/percentage of SIIFEKL (CFSE_low_) or OVA pulsed (CFSE_medium_) cells. To calculate the specific lysis percentage, the calculated ratio was normalized to control non-vaccinated mice: % specific lysis = (1 − (ratio of control recipient/ratio from vaccinated recipient) × 100).

### 2.8. Statistical Analysis

Statistical significance was calculated using Graphpad Prism 6 (GraphPad software Inc, San Diego, CA, USA). For pairwise comparisons, an unpaired one-tailed Student’s *t*-test was performed. Differences were considered significant if *p* < 0.05. Graphpad Prism 8 was also used to draw the graphical representations after importing the raw data from Microsoft Excel (Microsoft, Redmond, WA, USA).

## 3. Results

To evaluate and quantify cross-presentation by BMDC, we used a commercially available antibody (ab 25-D1.16) [10] that recognizes the peptide of the amino-acidic sequence SIINFEKL when presented in the MHC-I context of C57BL/6 mice cells. We then evaluated the level of cross-presentation in BMDC incubated with OVA +/− CDG (5 µg/mL) at several time points. Although the presence of CDG had no significant impact on cross-presentation by BMDC at early timepoints (4 and 8 h), CDG-treated BMDC showed a significantly higher cross-presentation compared to those treated with OVA alone at 24 h (Figure 1A).

We previously showed that the cross-presentation promoted by another CDN, CDA, uses the cytosolic pathway for antigen processing, thereby being cathepsin-independent and proteasome-dependent [6]. To determine whether CDG-mediated cross-presentation in BMDC is cathepsin- or proteasome-dependent, we evaluated CDG-mediated cross-presentation in the presence of the cathepsin inhibitor leupeptin and the proteasomal inhibitors lactacystin and MG-132. We observed that CDG promotes the cross-presentation of the antigen in a cathepsin-independent and proteasomal-dependent manner (Figure 1B,C). To further analyze the factors involved in CDG-mediated cross-presentation of soluble antigen, we measured the cross-presentation in BMDC that were generated from mice that lack the tumor necrosis factor (TNF) receptor (TNFR1a/b^−/−^) or type I IFN receptor (IFNAR1^−/−^). We found that the CDG-dependent increment in cross-presentation, similar to CDA, is IFN-α/β-dependent and TNF-α-independent (Figure 1D). We then compared the levels of cross-presentation attained by both adjuvants after 24 h. We observed that the lack of type I IFN signaling equally affected the cross-presentation capabilities induced by both adjuvants, whereas the lack of TNF signaling did not result in any reduction of the cross-presentation levels (Appendix A).

For CD8 T cell activation by DC, the engagement of the T cell receptor with the peptide/MHC-I complex on DC is necessary and represents the first activation signal. The second activation signal is induced through the binding of CD28 and CD40 ligand on T cells with their respective co-stimulatory molecules CD80/CD86 and CD40 on DC [11]. We previously observed that the presence of antigen deeply affects the activation of gene expression elicited by CDA in BMDC, since the cell activation did not occur in the absence of antigen [6]. To test if this feature was shared among CDNs and in particular between CDA and CDG, we assessed the levels of expression of the co-stimulatory molecules CD40 and CD86 on BMDC in the presence and absence of OVA. We observed that CDG (similar to other antigen-independent immunostimulators, e.g., LPS) can trigger the expression of the co-stimulatory molecules CD86 and CD40 in the absence of the antigenic protein OVA (Figure 2A,B). Interestingly, this CDG feature is also dependent on type I IFN signaling (Figure 2C,D).

Based on our previous works and different reports, it is known that the absence of type I IFN signaling results in underdeveloped humoral immunity in adults and neonates [12,13,14]. Thus, we measured the number and activation status of B cells after immunization with OVA + CDG/CDA or OVA alone in WT and IFNAR1^−/−^ mice. We observed a significant reduction in the total number of cells in the (draining) inguinal lymph node including the number of B cells (CD19^+^ cells) in IFNAR1^−/−^ mice with respect to WT mice when immunized with CDN as adjuvants (Figure 3A,B).

It is worthy of note that, in addition to reduced numbers, this subpopulation also showed a poorer activation status, as determined by the lower expression of CD69 and CD86 (Figure 3C,D). As expected, the decrease in B cell survival and activation resulted in diminished antigen-specific serum IgG titers after vaccination when using CDN (Appendix A) as an adjuvant in IFNAR1^−/−^ with respect to WT mice.

Next, we assessed whether CDG-induced cross-presentation can promote cross-priming in vivo. To this end, we performed a passive transfer of CFSE-labeled OT-I CD8^+^ T cells into congenic recipient mice. After 24 h we immunized the recipient mice. Then, 72 h following immunization, we isolated the spleen and LN and measured the proliferation of OT-I CD8^+^ T cells by flow cytometry (as CFSE dimming upon cell division) [9]. We also observed that the cross-priming capability is significantly reduced in IFNAR1^−/−^ mice when vaccinated with CDG (Figure 4A,B).

To investigate the in vivo relevance of the observed effect, we immunized mice according to the scheme in Figure 5A. We then performed an in vivo CTL assay to assess the cytotoxic capability of the activated CD8^+^ T cells. To this end, OVA- or SIINFEKL-pulsed CFSE-labeled splenocytes were transferred into congenic vaccinated mice.

**Figure 5 pharmaceutics-14-02821-f005:**
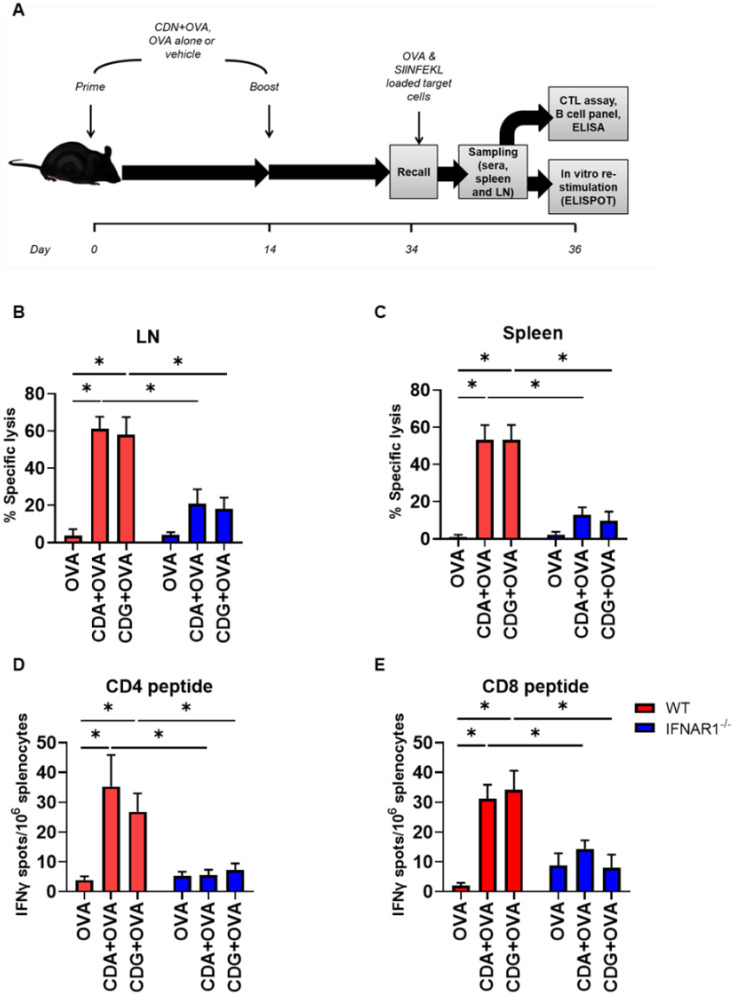
Type I IFN is necessary for CDN-mediated OVA-specific CTL generation in vivo. (**A**) CTL capacity to lyse SIINFEKL-pulsed target cells was assessed in vivo in vaccinated WT and IFNAR1^−/−^ mice. The bar graphs show the frequency of SIINFEKL-specific lysis measured in the (**B**) inguinal lymph nodes and (**C**) spleen of vaccinated mice. IFNγ-secreting splenocytes were quantified in the presence of (**D**) the CD4 peptide and (**E**) the CD8 peptide. Error bars indicate SEM. Statistical significance was calculated using a one-tailed Student’s *t*-test. * indicates *p* < 0.05. Data shown are from two independent experiments (WT, n = 7–8; IFNAR1^−/−^, n = 6–9).

These SIINFEKL-loaded CFSE-labeled splenocytes acted as a target and therefore can be killed by resident SIINFEKL-specific CD8^+^ CTL, which would diminish the CFSE signal. We observed that IFNAR1^−/−^ mice were significantly deficient in killing SIINFEKL-loaded target cells in both the LN (Figure 5B) and spleen (Figure 5C). Interestingly, significant differences were also observed in the LN and spleen when target cells were primed with whole OVA protein when using CDN (Appendix A) as an adjuvant in IFNAR1^−/−^ with respect to WT mice.

We then used ELISPOT to corroborate if the IFN-γ secretion was altered by the lack of type I IFN signaling, when mice were vaccinated with CDG + OVA. To this end, we measured IFN-γ secretion after immunization and re-stimulation (recall) according to the diagram displayed in Figure 5A. Since the re-stimulation of splenocytes was done with both MHC class I- and class II-restricted OVA immunodominant peptides, we were able to evaluate the secretion of IFN-γ by CD8 or CD4 T cells specifically. We observed that the secretion of IFN-γ by CD8^+^ T cells (which is a CTL distinctive feature) and by CD4^+^ T cells (a landmark of a Th1 immune response) were significantly impaired in IFNAR1^−/−^ mice vaccinated with CDG + OVA as compared with the CDG + OVA vaccinated WT counterparts (Figure 5D,E). Similar results were observed in CDA + OVA vaccinated animals (Figure 5D,E).

## 4. Discussion

We previously demonstrated that when CDA is used as an adjuvant, it only promotes cross-presentation, cross-priming and subsequent CTL responses when a functional type I IFN signaling is in place in APC. Nevertheless, these conditions were not proven to be necessary for CDG adjuvanticity until now. In this regard, whereas the pivotal importance of IFN-α/β for CDN adjuvanted immune responses was widely acknowledged [6,15,16], it has been controversially claimed that type I IFN is nonessential for CDN adjuvanticity [17,18]. To solve this conundrum, we showed strong evidence supporting the critical role of type I IFN for CDG-mediated immune responses. One of these facts is that CDG activates BMDC through the up-regulation of CD40/86 (signal II for T cell activation) only in the presence of functional type I IFN signaling. Signal II is essential to provide co-stimulation for the successful activation, survival and proliferation of T cells. Considering that the absence of the co-stimulatory signal renders T cells anergic, this hypo-responsiveness would have negative effects on the immune response mounted in IFNAR1^−/−^ animals. Nevertheless, as we showed previously for CDA, this is not a property shared by all STING agonists [6]. Among the responses downstream of the generation of type I IFN, the secretion of IFN-γ is a feature that determines the generation of functional CTL (among CD8^+^ T cells) and Th1 (among CD4 T cells). When we re-activated splenocytes with the OVA MHC class I and II restricted peptides in previously vaccinated animals, we found that the lack of type I IFN signaling impaired CDN-mediated secretion of IFN-γ. Interestingly, the abrogation of proliferation in response to signals 1 and 2 exerted by the lack of one cytokine signaling (signal 3) shows how important cytokine signaling is for a properly regulated immune response. Thus, we corroborated the importance of type I IFN for CDG-driven Th1 responses. In addition to the described limitations, the effects of a significantly reduced cross-presentation in APC must be considered. Therefore, in a consistent mode with what we previously found for CDA, we showed here that the lack of type I IFN signaling impairs the CDG-elicited cross-presentation, cross-priming and subsequent CTL generation. Moreover, we found that the lack of type I IFN not only affects the CTL response, but also the survival, maturation and expansion of B-lymphocytes, as well as the subsequent production of antigen-specific antibodies when CDA or CDG are used as adjuvants.

When a CDN is proposed as an adjuvant or immune modulator in a clinical setting, it is important to understand its mode of action. Among the arms of the immune response that were targeted by using CDN in clinical trials, its ability to generate prominent CTL responses is central [1]. When using a soluble antigen, CTL generation is a result of antigen cross-presentation which relies on two main cross-presentation pathways: the cytosolic and the vacuolar. We previously demonstrated that CDA promotes the cytosolic cross-presentation pathway. We corroborated that CDG-mediated cross-presentation is also proteasome-dependent and cathepsin-independent, thus clarifying the mode of CTL generation for the two most relevant CDN candidates in vaccine development. Interestingly, CDG-mediated cross-presentation, similar to CDA, is TNF signaling independent.

The emerging knowledge on the underlying mechanisms behind CDN adjuvanticity presented here could be applied for improved vaccination strategies when these molecules are incorporated in the formulation. Taken together, this work elucidated the importance of IFN-α/β signaling in CDG-mediated adjuvanticity. Besides T cell-mediated cellular immunity, the development and activation of B-lymphocytes for the induction of humoral immunity are also of key importance for the design of new vaccines. New technologies for rapid vaccine development played a crucial role in containing the COVID-19 pandemic and the economic recovery. The use of new adjuvants such as CDN, acting also when administered by the mucosal route, can greatly improve protection against infection, reduce horizontal transmission and extend immune memory, thereby reducing the number of doses in both conventional and pandemic scenarios.

## Figures and Tables

**Figure 1 pharmaceutics-14-02821-f001:**
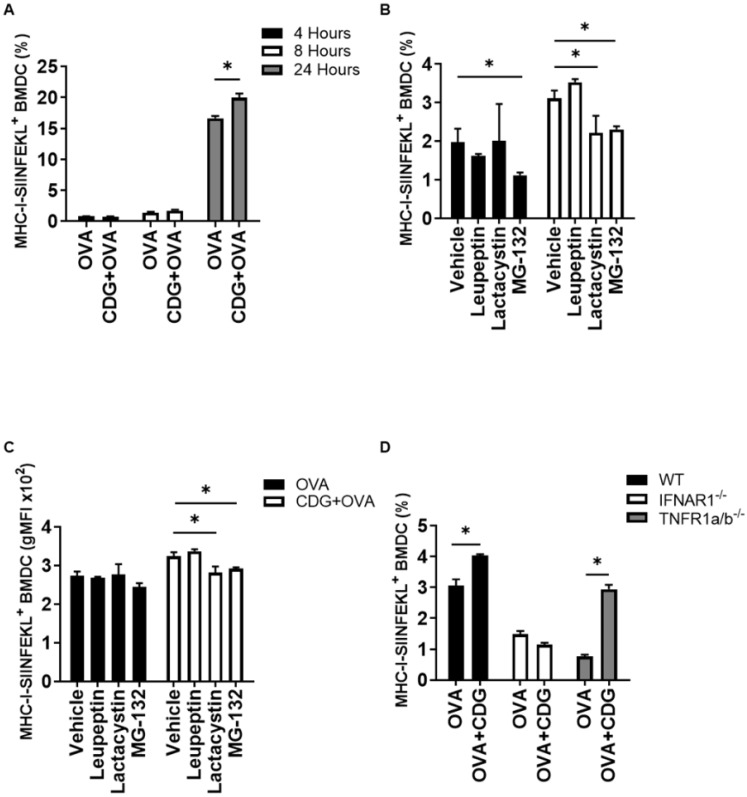
Cross-presentation of the OVA immunodominant peptide SIINFEKL in BMDC pulsed with OVA +/− CDG. (**A**) WT BMDC were treated with OVA +/− CDG for 4, 8 and 24 h. The frequency of cross-presenting BMDC was quantified using flow cytometry. (**B**) Frequency or (**C**) geometric mean of the fluorescence intensity (gMFI) of cross-presenting BMDC were quantified in the presence or absence of inhibitors of the cytosolic pathway (MG-132 or lactacystin) and an inhibitor of the vacuolar pathway (leupeptin). (**D**) Cross-presentation of BMDC from IFNAR1^−/−^ and TNFR1a/b−/− mice was compared to WT counterparts in the presence or absence of CDG. Error bars indicate SEM. Statistical significance was calculated using a one-tailed Student’s *t*-test. * indicates *p* < 0.05. The results are representative of two to three independent experiments.

**Figure 2 pharmaceutics-14-02821-f002:**
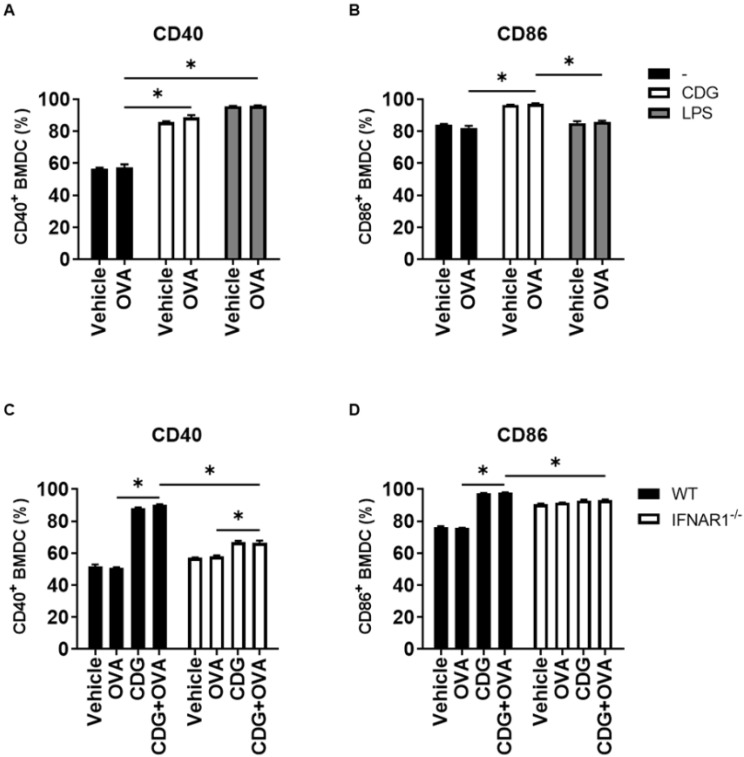
CDG-mediated upregulation of the costimulatory molecules CD40 and CD86 is type I IFN dependent. WT BMDC were treated with OVA +/− CDG or LPS for 24 h. The frequency of (**A**) CD40+ and (**B**) CD86+ BMDC were quantified using flow cytometry. WT and IFNAR1^−/−^ BMDC were treated with OVA, CDG or OVA + CDG and the frequency of (**C**) CD40+ and (**D**) CD86+ BMDC were quantified using flow cytometry. Error bars indicate SEM. Statistical significance was calculated using a one-tailed Student’s *t*-test. * indicates *p* < 0.05. The results are representative of two independent experiments.

**Figure 3 pharmaceutics-14-02821-f003:**
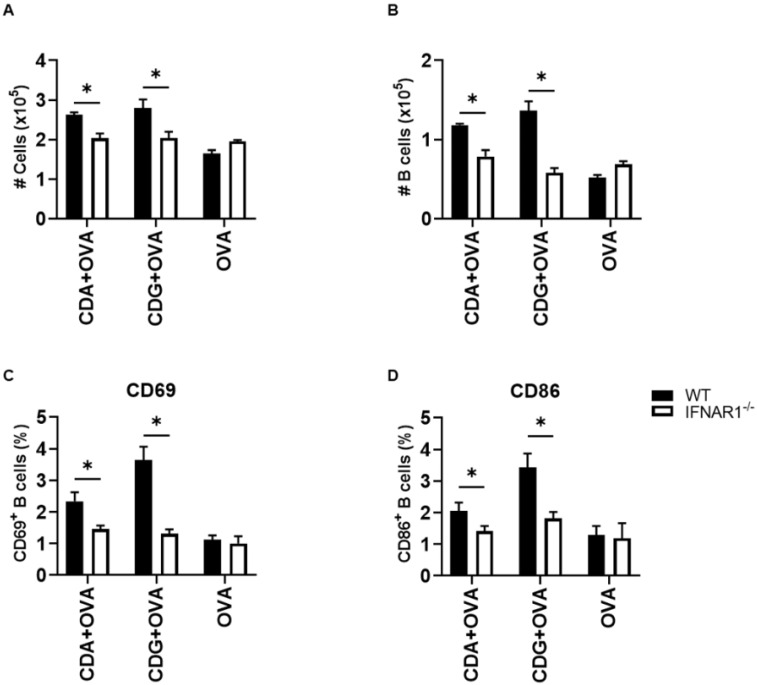
CDG-mediated expansion and activation of B cells are type I IFN dependent. B cell expansion and activation were assessed in vaccinated WT and IFNAR1^−/−^ mice following rechallenge with OVA-pulsed target cells. The bar graphs show the absolute count of (**A**) total cells and (**B**) B cells and the frequency of activated (**C**) CD69^+^ and (**D**) CD86^+^ B cells in the inguinal lymph node of vaccinated mice. Error bars indicate SEM. Statistical significance was calculated using one-tailed Student’s *t*-test. * indicates *p* < 0.05. The results are representative of two independent experiments (WT, n = 4; IFNAR1^−/−^, n = 3–4).

**Figure 4 pharmaceutics-14-02821-f004:**
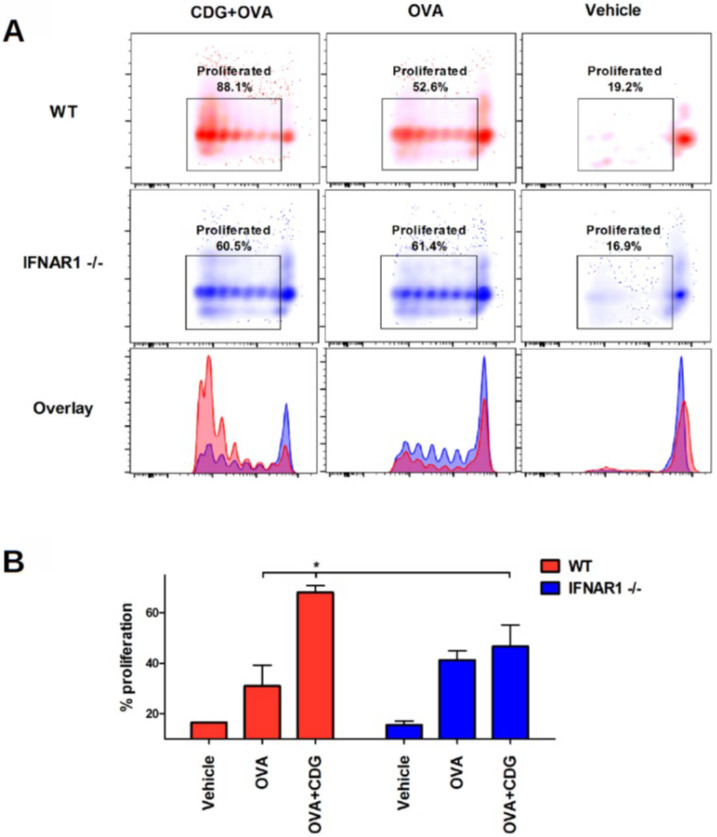
Type I IFN is necessary for CDG-mediated CTL proliferation in vivo. CFSE-labelled CD8 OT-I cells were passively transferred into WT and IFNAR1^−/−^ mice. 16 h later, mice were vaccinated with CDG + OVA, OVA or vehicle. The dilution of CFSE intensity was measured after 3 d using flow cytometry. (**A**) representative plots and overlays. (**B**) frequency of proliferated CD8 OT-I cells. Error bars indicate SEM. Statistical significance was calculated using one-tailed Student’s *t*-test. * indicates *p* < 0.05. Data shown are representative of three independent experiments (WT, n = 3–4; IFNAR1^−/−^, n = 3–4).

## Data Availability

Not applicable.

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
