# Peer review of "IFN-α/β Signaling Is Required for CDG-Mediated CTL Generation and B Lymphocyte Activation"

_pharmaceutics, 2022, doi:10.3390/pharmaceutics14122821_

Round 1

Reviewer 1 Report

The manuscript  IFN-α/β signaling is not dispensable for CDG-mediated CTL generation and B lymphocyte activation is an interesting work ; however the title is ambiguous, it should be reconsidered.

There is consistent evidence to suggest that IFN-α/β (Type I IFN) is crucial for CDG-mediated B cell activation and the authors of this paper have determined the role of type I IFN signaling for CDA-mediated enhancement of immunogenicity and the hypothesis in this paper suggests that it is required for CTL induction. It is interesting to note that the effect of this adjuvant is intended to be identified in a model of response to soluble antigen, when a particulate antigen should be considered. It is difficult to consider that the objectives and the proposed hypothesis have been met, v.gr. Figure 1 identifies that there are no differences between the MHC 1 response between WT and IFNKO, in response to CDGA-OVA.

Another of the proposals put forward by the authors is that the mode of action of type I IFN signaling is disclosed in CDG-mediated cross-presentation and subsequent CTL generation. Unfortunately, it is not clearly identified that the effect of immune regulation is dependent on CDG.

Unfortunately the discussion is more focused on previous studies. It is suggested that it should be better focused on present results

Author Response

Reviewer 1

The manuscript  IFN-α/β signaling is not dispensable for CDG-mediated CTL generation and B lymphocyte activation is an interesting work; however the title is ambiguous, it should be reconsidered.

We thank the reviewer for their suggestion. We wanted to emphasize the role of type I interferon (IFN) signaling in CDG-mediated CTL generation and B cell activation, however, we agree with the reviewer that the current title might be ambiguous. Therefore, we changed the title of the manuscript to “IFN-α/β signaling is required for CDG-mediated CTL generation and B lymphocyte activation”, which was the original title of the paper before internal editing.

There is consistent evidence to suggest that IFN-α/β (Type I IFN) is crucial for CDG-mediated B cell activation and the authors of this paper have determined the role of type I IFN signaling for CDA-mediated enhancement of immunogenicity and the hypothesis in this paper suggests that it is required for CTL induction. It is interesting to note that the effect of this adjuvant is intended to be identified in a model of response to soluble antigen, when a particulate antigen should be considered.

We kindly appreciated the suggestion and we certainly will use particles in future research, to see if these can overcome Type I IFN dependency. It is also important to consider that we want to show adjuvant properties that are independent of the nature of the antigen (soluble or particulate). One very desirable feature for an adjuvant is the capability to elicit a Type I IFN dominated immune response (T helper I, TH1) when used with soluble antigens, this means not only that the adjuvant favors cross-presentation but also that can be used in vaccines with lyophilized protein (antigen).

It is difficult to consider that the objectives and the proposed hypothesis have been met, v.gr. Figure 1 identifies that there are no differences between the MHC 1 response between WT and IFNKO, in response to CDGA-OVA.

We appreciate the reviewer’s comment, however, Figure 1D clearly shows that CDG-mediated cross-presentation of the OVA immunodominant peptide SIINFEKL by BMDC is type I IFN dependent as it was abolished in the IFNAR1-/- BMDC.

Another of the proposals put forward by the authors is that the mode of action of type I IFN signaling is disclosed in CDG-mediated cross-presentation and subsequent CTL generation. Unfortunately, it is not clearly identified that the effect of immune regulation is dependent on CDG.

We kindly appreciated the comment pointing at the mode of action of Type I IFN, but indeed we only claimed that we disclosed the importance on Type I IFN on the CDG-mediated cross-presentation, further cross-priming and subsequent CTL generation. We never tried to specifically investigate details of Type I IFN signaling other than the effects of the presence/absence of Type I IFN signaling on CDG immunogenicity.

Unfortunately the discussion is more focused on previous studies. It is suggested that it should be better focused on present results.

Although the Discussion section is intended to discuss our results, when doing so, it is not possible to ignore the previous knowledge about the topic, in the same way that experiments have their own positive and negative controls, and many times these are Gold Standards investigated previously by other researchers.

Given the level of detail that we displayed in the figures and their legends, in the methods section and explicitly argued in the introduction and the results sections, we found that we hardly can improve the discussion section by adding more details, but on the contrary, we are afraid to sound extremely redundant (and therefore obscure or boring) for the Pharmaceutics readers.

Reviewer 2 Report

This paper describes characteristics of CDG as a potential adjuvant in immunization. Previously, the authors reported that type I IFN is essential for CDN-mediated cross-presentation and CTL generation, demonstrating the case of CDA as an example. In this paper, similar to the case of CDA, CDG triggered cross-presentation and CTL generation in IFN signaling dependent manner. In addition, mode of action for CDG-mediated cross-presentation was explored. Overall, this reviewer recommend publication of this paper in Pharmaceutics after minor revision. Comments for revision is as follows: Both CDA and CDG share a similar mode of action for cross-presentation and CTL generation. Are they synergistic or competitive? Compare the two adjuvants and add it to discussion.

Author Response

Reviewer 2

This paper describes characteristics of CDG as a potential adjuvant in immunization. Previously, the authors reported that type I IFN is essential for CDN-mediated cross-presentation and CTL generation, demonstrating the case of CDA as an example. In this paper, similar to the case of CDA, CDG triggered cross-presentation and CTL generation in IFN signaling-dependent manner. In addition, mode of action for CDG-mediated cross-presentation was explored. Overall, this reviewer recommend publication of this paper in Pharmaceutics after minor revision.

We thank the reviewer for their critical and constructive review of our manuscript and the appreciation of our findings.

Comments for revision is as follows: Both CDA and CDG share a similar mode of action for cross-presentation and CTL generation. Are they synergistic or competitive? Compare the two adjuvants and add it to discussion.

Figure (please see attached files)

Figure 1: Cross-presentation of the OVA immunodominant peptide SIINFEKL in BMDC pulsed with OVA+/- CDG and CDA

Indeed, we believe that both CDA and CDG share a similar mode of action, which has raised the question of whether CDA and CDG are synergistic. Therefore, we have performed an experiment in which we investigated the effect of combining CDA and CDG on the cross-presentation capacity of BMDC. To this end, WT and IFNAR1-/- BMDC were incubated with OVA alone or in the presence of CDA, CDG or the combination of CDA and CDG. Although CDA+CDG has significantly enhanced the cross-presentation of OVA by BMDC, it was not better than CDA or CDG alone (Figure 1). Since both CDA and CDG enhance cross-presentation in a type I IFN-dependent manner, the combination of CDA and CDG failed to enhance cross-presentation in IFNAR1-/- BMDC (Figure 1).

Based on these results, we have concluded that despite their similar mode of action, CDA and CDG are not synergistic.

Reviewer 3 Report

In this manuscript, the authors demonstrate the need of type I IFN signaling induced by CDG-mediated B cell activation.  When working with vaccine adjuvants, it is always a problem to unravel the mode of action and this work sheds light on CDN-type adjuvants. The use of OVA as a model is pertinent, but I would prefer to see a study with other vaccine antigens of interest. Anyway, the manuscript is clear and well-written, within its proposal.

Author Response

Reviewer 3

In this manuscript, the authors demonstrate the need of type I IFN signaling induced by CDG-mediated B cell activation.  When working with vaccine adjuvants, it is always a problem to unravel the mode of action and this work sheds light on CDN-type adjuvants. The use of OVA as a model is pertinent, but I would prefer to see a study with other vaccine antigens of interest. Anyway, the manuscript is clear and well-written, within its proposal.

We thank the reviewer for their critical and constructive review of our manuscript and the appreciation of our findings.